# Protecting Chiller Systems from Cyberattack Using a Systems Thinking Approach

**Shaharyar Khan * and Stuart Madnick**

Sloan School of Management, Massachusetts Institute of Technology, Cambridge, MA 02139, USA
* Correspondence: shkhan@mit.edu

**Abstract:** Recent world events and geopolitics have brought the vulnerability of critical infrastructure to cyberattacks to the forefront. While there has been considerable attention to attacks on Information Technology (IT) systems, such as data theft and ransomware, the vulnerabilities and dangers posed by industrial control systems (ICS) have received significantly less attention. What is very different is that industrial control systems can be made to do things that could destroy equipment or even harm people. For example, in 2021 the US encountered a cyberattack on a water treatment plant in Florida that could have resulted in serious injuries or even death. These risks are based on the unique physical characteristics of these industrial systems. In this paper, we present a holistic, integrated safety and security analysis, we call Cybersafety, based on the STAMP (System-Theoretic Accident Model and Processes) framework, for one such industrial system—an industrial chiller plant—as an example. In this analysis, we identify vulnerabilities emerging from interactions between technology, operator actions as well as organizational structure, and provide recommendations to mitigate resulting loss scenarios in a systematic manner.

**Keywords:** cybersecurity; cybersafety; stamp; STPA; system security; industrial chillers





## 1. Introduction

While there has been considerable attention to attacks on Information Technology (IT) systems, such as data theft and ransomware, the vulnerabilities and dangers posed by industrial control systems (ICS) have received significantly less attention. Events such as the cyberattacks on the Ukrainian power grid, as well as attacks on oil and gas plants and nuclear facilities in Saudi Arabia and Iran, respectively, have demonstrated not only the capability but also the willingness of nation-states and advanced cyber adversaries to disrupt and/or cause damage to an adversary's critical infrastructure [1]. Part of the reason for the lack of attention to cyberattacks on ICS is because of an underlying assumption that the control systems (that operate the pumps, valves and machines) are isolated from the public internet. While true in theory, the increased digitalization, operational integration and automation has led to increased complexity of the systems at the cost of degraded security. Hence, even remote industries which use ICS for core functionality (such as maritime sector, etc.) that were hitherto considered safe from cyberattacks are under threat and have noted a significant increase in cyber breaches [2].

ICS can fall victim to the same kinds of threats experienced by IT systems, such as disruption of operations. However, there is an important difference; ICS can be made to do things that could destroy equipment or even harm people. For example, recently, the US encountered a cyberattack on a water treatment plant in Florida that could have potentially resulted in serious injuries or even death. These risks arise due to the unique physical characteristics of industrial control systems, which require a detailed and serious analysis of not only the security of information exchanges inside these systems but also the safety-related consequences that can emerge as a result of malicious actions.

In this paper, we use the chiller plant as an illustrative example of an archetypal ICS that uses large, specialized industrial equipment that could be the target of a cyberattack. Chiller plants are sometimes ignored as potential security targets because they are classified as non-critical to the business process; however, it is important to note that (1) chiller plants along with industrial boilers are embedded within every aspect of our lives—from large commercial and office buildings, data-centers, hospitals, college campuses to ice rinks, shopping malls and grocery stores—and, (2) unsafe operation can result in catastrophic consequences. For instance, although not cyber-related, the 1997 chiller accident at Los Alamos National Labs cost $3.2 million in damages [3] due to flooding of a facility subbasement that stored weapons-grade radiological sources. In addition to the direct first-order effects on life and/or property, an attack on a chiller system could have significant nth-order effects. For instance, an attack on a chiller system that is required to maintain environmental control at a data-center could potentially cause service interruption or disruption; the chiller system could in fact act as a proxy for an attack on the data center or other network facilities.

In the context of cybersecurity, the traditional approach to protecting such systems is to undertake a risk-based, technical perspective that is biased by information security concerns [4] and focused on security of individual components, assuming direct, linear causality leading to cyber incidents. However, important differences exist between cyber-physical and traditional IT systems, that make such a narrow approach largely impotent in the face of targeted attacks [5]—underscoring the need for a systems perspective of the joint security and safety problem.

The primary contribution of this work is to present an integrated safety and security model for an industrial control system application, that we call Cybersafety [6], based on the STAMP STPA-Sec method. While some authors have performed cybersecurity analyses using the STAMP framework in the past, we believe those are essentially safety analyses containing cybersecurity-related causal factors. What is missing is an integrated safety/security model that analyzes a complex industrial process from both lenses—safety and security. To that end, the objective of this paper is to demonstrate that an integrated safety/cybersecurity analysis is feasible on a real industrial control system and highlight refinements to the Cybersafety [6] method that enable such an analyses.

One example of the types of things we identify in this analysis is how the functionality to remotely update critical frequency settings for the variable frequency drive (VFD) controlling the chiller compressor (common in most industrial plant environments) can be the target of an attack to cause permanent damage to the chiller compressor. While this may not be 'news' for most plant operators, we go deeper to understand systemic factors including structural and process model flaws that enable existence of such vulnerabilities and offer recommendations on the entire system (people, processes, technology and organization) that can be leveraged to prevent such losses from such weaknesses.

We begin by reviewing literature about the application of systems theory to cyber-security in Section 2 along with an identification of gaps in current approaches. Next, a brief overview about the Cybersafety method along with a system-level description of the chiller plant and more specifically, operation of a centrifugal chiller is provided in Section 3. Section 4 contains the bulk of the analysis employing the Cybersafety method. A discussion of the results, along with some proposed mitigation requirements is provided in Section 5 followed with some final remarks in Section 6.

## 2. Literature Review

The traditional method to protect against cyberattacks in the IT world is to take a static, risk-based approach. This includes identification of threat actors and threat events and their relevance to the organization, followed by identification of vulnerabilities in the system (based on software configuration, system architecture, hardware/asset inventory, etc.), and likelihood of exploitation based on protection barriers in the system (firewalls, access control, intrusion detection, etc.), and finally, a determination of probable impact of a

loss event. Combined together, this enables security practitioners to calculate risk as a combination of likelihood and impact, which is then used to inform security decisions.

When it comes to cyber-physical systems, however, a cyberattack has the potential to impact system safety and cause actual physical damage. As a result, a number of hazard analysis frameworks, traditionally employed for safety analyses have been adapted for security analyses. For instance, Schmittner et al. [7] proposed extending FMEA to include vulnerabilities, threat agents and threat modes as inputs for determining failure causes—with the new approach known as Failure Mode, Vulnerabilities and Effects Analysis (FMVEA). While well-suited for evaluating individual component failures, it lacks as a holistic safety/security methodology because it does not consider failures due to component interactions [8].

Similarly, Steiner and Liggesmeyer [9] proposed an extension of another well-known hazard analysis method, i.e., Fault-Tree Analysis (FTA) [10] by modeling attacker's intentions in the analysis; the extended FTA method is known as the Extended Tree Analysis (ELT) [11]. This method is also limited in its analysis of human factors, organizational and extra-organizational factors [12]. Another well-recognized systems-based hazard analysis methodology is the Hazard and Operability (HAZOP) Analysis method [8] which lies in between FMEA and FTA. Researchers have tried to formalize HAZOP to achieve objective and quantifiable results, "but all approaches to quantify results have led back to the use of FTA" [13].

Each of these preceding hazard analyses methods is based on the practice of analytic reduction—an approach to manage complexity—where it is assumed that by breaking the system into smaller components, examining and analyzing each component in detail in isolation and then combining the results, the properties or behaviors of the system can be understood [14]. Two implicit assumptions are made here (1) there are no indirect interactions between components and (2) behavior of the system can be modeled as separate linear events where each event is the direct result of the preceding event. These assumptions, while sufficient for simple systems, do not hold for complex, sociotechnical systems where the components interact in many indirect ways.

An alternative to performing joint analysis of safety and security using extended versions of traditional hazard analysis methods (such as FTA [11], FMEA [7] etc.), is to use the perspective of modeling using systems theory.

Leveson [14,15] developed a framework to understand causes of accidents using systems theory known as STAMP (System-Theoretic Accident Model and Processes). STAMP is a framework that treats accidents as a 'dynamic control problem' emerging from violation of safety constraints rather than a 'reliability problem' aimed at preventing component failures [14]. While STAMP is an accident-causality framework or set of assumptions about how and why accidents happen, several analytical methods have been developed based on the STAMP framework such as STPA, CAST, STPA-Sec, etc.

Laracy & Levenson [16] proposed extending STAMP to analyze security of critical infrastructure (known as STAMP-Sec) by adding a System Dynamics modeling component to a traditional STAMP analysis. Using the pre-9/11 Air Transportation System as an example, this work [16] proposes that system dynamics modeling could help explain the why or behavioral dynamics of why inadequate control could occur in a complex system. Although a detailed analytical example is not provided, Laracy & Levenson 's work [16] identifies differences in language between safety and security analyses (hazards vs. threats, safety constraints vs. security constraints).

Laracy & Levenson [16] are followed by Young & Leveson [17] who presented STPA-Sec in a concept paper as a methodology to perform integrated safety and security analysis using systems theory. Being a concept paper, they stop short of providing a detailed example of application of the STPA-Sec on a real system.

Meanwhile, Hamid & Madnick [18] analyzed the TJX cyberattack (discovered in 2007, the cyberattack targeting TJX Corporations was at the time the largest breach of consumer data impacting 94 million records) using the STAMP framework while Nourian

& Madnick [19] analyzed the Stuxnet attack using the STAMP framework. However, both these analyses focused on events that had already occurred—they were not forward-looking analyses.

In forward-looking analyses based on STAMP, Logan et al. [20] analyzed an autonomous space system using the STPA-Sec method to elicit system security requirements while Martin et al. [21] applied STPA-Sec to study security considerations in aerial refueling. Meanwhile, Khan & Madnick [6], presented the STAMP-based Cybersafety analysis of a gas-turbine power plant. In each of these analyses, the focus is on identifying safety issues emanating from security scenarios.

Others have proposed extensions to STPA with threat models (such as STRIDE (STRIDE is a mnemonic that stands for Spoofing, Tampering, Repudiation, Information Disclosure, Denial of Service, and Elevation of privilege) [22]). This approach provides a more formal method of cataloging scenarios, providing a common language that is well understood by security practitioners. Meanwhile, Friedberg et al. [13] presented an analytical methodology that combines safety and security analysis, known as STPA-SafeSec where the main contribution was mapping STPA's abstract functional control diagram to a physical component diagram and using the physical component level to perform a vulnerability/security assessment. While the addition of a physical component diagram makes the analysis more concrete and enables traditional security analysis techniques to be reconciled with STAMP, it fails to capture the essence of a STAMP analysis, i.e., to identify vulnerabilities that exist due to violation of security constraints and component interactions at the functional level.

Evaluating the above-mentioned STAMP-based analyses, we discover that they attempt to identify how the safety control structure for the system, can be manipulated by an ill-intentioned adversary to cause safety issues. However, when analyzing industrial control systems, we believe there is a need to analyze security controls in parallel with safety controls—which is a gap in the current STPA-Sec/Cybersafety analysis methodologies.

To illustrate this point, consider a thermostat in a room. The traditional STAMP-based analyses evaluate safety controls that are in place to ensure the thermostat maintains temperature within certain defined limits and identify how the various controls may fail under adversarial actions (such as deliberate loss of feedback, loss of control, etc.). What we are proposing is that in addition to safety controls, we must also include the security controls in the same functional control diagram and assign security responsibilities to each of the controllers. In the thermostat example, this would imply that in addition to maintaining temperature between safe limits, the thermostat would also be assigned a security constraint to ensure access control. This would ensure that not only the safety control structure is analyzed but the security control structure is also assessed in parallel. Our literature search did not reveal any such detailed published work where an integrated safety and security analysis of an industrial control system has been performed. This is an important contribution of this paper.

## 3. Cybersafety Method

The STAMP framework [14,15] asserts that accidents are a result of loss of control. Here, the key idea is that a complex system consists of a number of interacting decision-makers or controllers at different abstraction levels of the system (technology, operational, management, regulatory), each trying to enforce certain safety and security constraints on the controlled processes within the system. System-level losses emerge as a result of violation of these constraints. In this paradigm, each controller-controlled process interaction is viewed as a feedback loop. Any disruption of information flows within this feedback loop results in degradation of the controlled process which ultimately causes the system to transition into an unsafe/insecure state. Security, like safety, is therefore, an emergent property of the system that cannot be achieved by securing individual components; rather it requires a holistic view and understanding of the system where each component interaction is understood in the context of the entire system—people, processes, technology

and organization. The STAMP-based STPA method is described in detail in the STPA handbook [14] while its application to a real-world industrial control system cybersecurity example is described as the Cybersafety method by Khan & Madnick [6]. In this section, we describe the proposed improvements to the Cybersafety method to better elicit security requirements from the analysis.

The basic steps in the Cybersafety method are illustrated in Figure 1 [6]. Briefly, the method starts by defining the basis of the analysis which includes identifying the goal of the system, the most critical losses and system-level hazards that can result in those losses (if not adequately controlled) along with system-level constraints to prevent the losses. The next step is to identify controllers responsible for enforcing constraints on the processes (i.e., to control the hazards) and their interactions with one another—this results in the development of a hierarchical functional control structure.

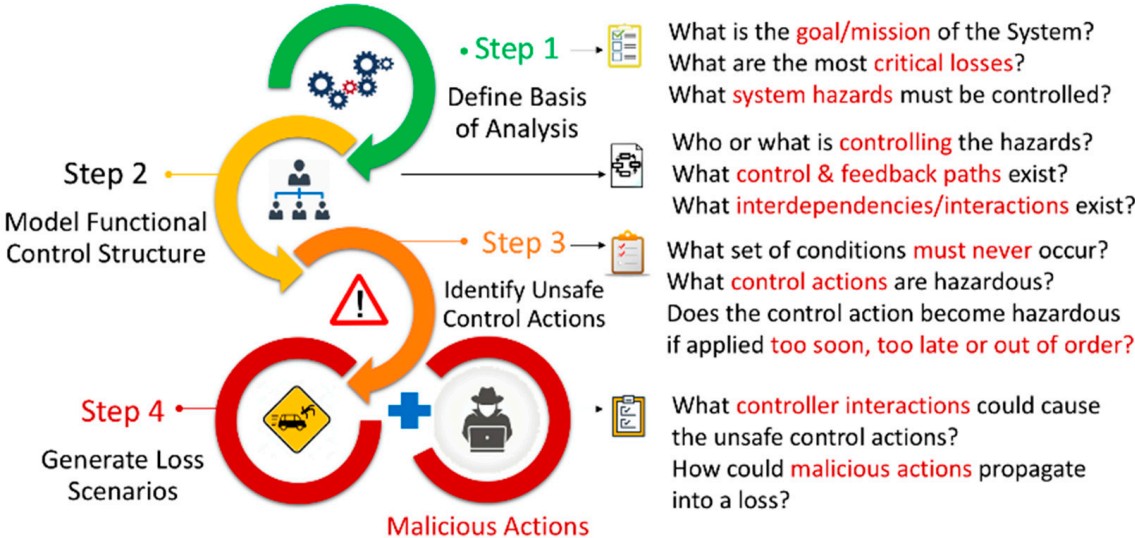

**Figure 1.** Overview of the Cybersafety method [6].

Next, each control action for each controller is evaluated in the context of the various system and environmental states that the system is subject to, in order to identify hazardous control actions. Finally, loss scenarios are generated by hypothesizing how the interaction between the controllers, flaws within the system and missing constraints can be leveraged by an attacker to cause system-level losses. Countermeasures are then derived to prevent the hazards from propagating into system-level losses.

We propose the following additions to the Cybersafety method to further focus the analysis on security issues and uncover systemic vulnerabilities:

In Step 1, system-level hazards should be phrased as system-level threats. This implies rewording the hazards in terms of attacker targets. This would help reduce the gap in terminology with traditional security analyses while focusing the attention of the analyst to 'think like an attacker'.

In Step 2, controllers enforcing security controls should be explicitly identified in addition to controllers enforcing safety constraints. In addition, for each controller, security responsibilities should also be explicitly identified in parallel to safety responsibilities.

In Step 3, insecure control actions for each controller should be identified in conjunction with unsafe control actions.

In Step 4, loss scenarios should encompass not only the failure of the control structure in enforcing safety constraints but also security constraints.

The addition of these steps would help to bridge the gap between STAMP-based security analyses and traditional approaches used by security analysts.

### 3.1. Description of the System

The chiller plant that is the subject of this work is assumed to be located inside an archetypal industrial facility. This industrial facility has upstream operations that include delivery of fuel (both natural gas and fuel oil) to the plant along with a tie-line connection to the local utility grid. In addition, the facility imports steam from the area's local powerplant to supplement its own production. The facility's downstream operations include distribution of electricity, steam and chilled water to the facility buildings.

In addition, the plant operates a 21 MW gas turbo-generator that provides electricity to the facility buildings; waste heat from the turbine is directed to a Heat Recovery Steam Generator (HRSG) to produce steam. The steam from the HRSG is supplemented with steam from other gas/oil-fired water-tube boilers and is used for driving steam-driven chillers as well as for heating. The combined output from the 6 steam-driven chillers is 21 kilotons. This chilled water supply is supplemented with 8 additional electric-driven chillers (with a combined capacity of 13 kilotons) to meet facility demand. The plant consists of a juxtaposition of various types of chillers (e.g., reciprocating, centrifugal, screw-driven, etc.), from different manufacturers and of different equipment ages which adds to the complexity of the system. For the purpose of this paper, we focus on electric-driven centrifugal chillers only.

### 3.2. Centrifugal Chillers

A chilled water system consists of a chiller or a combination of chillers, air-handling units (AHU), cooling towers as well as auxiliary equipment including pumps, valves, water purification system and piping as shown schematically in Figure 2. The chiller removes heat from a liquid via a vapor-compression cycle which consists of four main components: evaporator, compressor, condenser and expansion device. The basic operation can be described as follows.

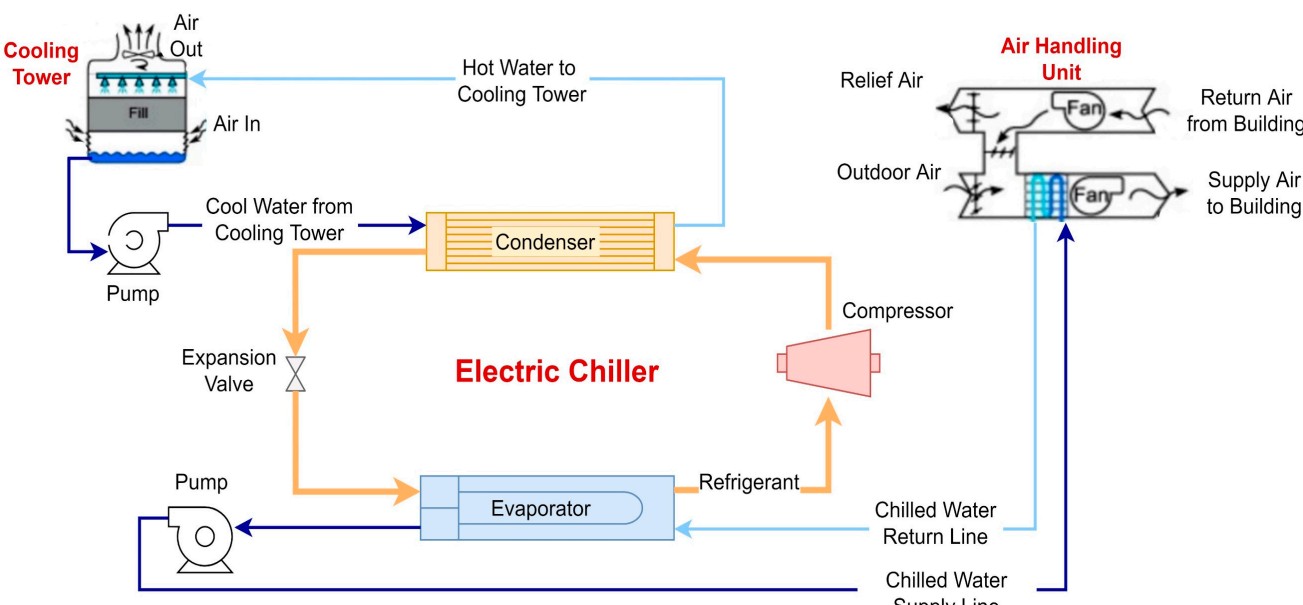

**Figure 2.** Schematic of Chiller Plant.

The refrigerant in the evaporator, absorbs heat from the chilled water return line (returned from the facility buildings), changing its state to superheated vapor. The temperature and pressure of this superheated refrigerant vapor is increased by the compressor which converts kinetic energy to pressure and pumps the vapor to the condenser. Here, cool condenser water returning from the cooling towers extracts heat from the refrigerant vapor converting it back to a high pressure, high temperature liquid. This high pressure,

high temperature liquid moves to the thermal expansion valve. The thermal expansion valve reduces the temperature of the liquid by reducing its pressure by passing the liquid refrigerant through a small adjustable orifice. The liquid refrigerant then again absorbs heat from the chilled water return line, turns to vapor and moves to the compressor; the cycle is then repeated [23].

Note that there are three independent fluid loops which function together to enable delivery of chilled water to the facility; (1) a closed loop water circuit that runs chilled water between the building AHUs and the evaporator, (2) a closed loop refrigerant circuit, which enables transfer of heat from the chilled water loop to the condenser water loop, and (3) an open water loop, which absorbs heat from the refrigerant and rejects it to the atmosphere via cooling towers. Each of these loops have pumps and valves which are operated by the chiller controller or supervisory controllers such as the Distributed Control System (DCS) or manually by the operator. The controllers take decisions based on information gathered by process sensors distributed across the plant.

## 4. Cybersafety Analysis

This section provides an overview of the application of the Cybersafety method to the chiller plant. It is divided into subsections where each subsection represents one step in the basic Cybersafety diagram presented in Figure 1.

### 4.1. Define Basis of Analysis

Being a top-down, consequence-driven method, the first step in the Cybersafety method establishes the system boundaries by defining the goal/primary mission of the target system (i.e., the Chiller Plant) using the system problem statement as follows:

**(A system) TO:** Provide chilled water for facility building temperature control, reliably, efficiently and safely;

**BY (how):** Controlling refrigeration cycle (to attain desired chilled water temperature set-point), managing flow of chilled water to facility buildings and rejecting waste heat;

**USING (what):** Control hardware/software including chiller controller, DCS, EMS, pumps and valves.

Note that the system problem statement highlights three critical functions identified—controlling chiller capacity, managing chilled water distribution and rejecting waste heat to the environment—that enable the system to achieve its primary value function.

Next, with the boundaries of the target system established, unacceptable system-level losses, system-level hazards and constraints are determined as itemized in Table 1. From the perspective of STAMP [14], there is a clear distinction between system-level losses and hazards where losses are unacceptable conditions from the mission owner/primary stakeholder's perspective vs. system-level hazards are those system-states which if not controlled, would result in losses. The goal of the analysis is to establish constraints on the system that prevent the hazards from translating into losses.

Traditionally, in a STAMP analysis, hazards are identified in terms of the system. This is deliberate because it shifts the focus away from components to system conditions that must be controlled. Second, it aims to strategically shift the goal of the analysis from one that focuses on protecting against external threats (which continuously change over time) to one that is aimed at uncovering internal weaknesses. However, for traditional security analysts it is of paramount importance to prioritize their defenses against the most credible threats to the system. Therefore, we propose identifying system-level threats that can lead to the unacceptable losses. Table 1 identifies both system-level hazards as well as system-level threats that can lead to unacceptable losses. By changing this perspective, we are able to strategically align the analysis with traditional security analysis terminology. Note that this same approach is proposed by Laracy & Levenson [16] as well in describing the STAMP-Sec approach for protecting critical infrastructure. In subsequent research, we would explore how the defender's threat assessment (including understanding of adversary's motives,

capabilities and targets) could be utilized to prioritize and reduce scope at this early stage of the analysis.

**Table 1.** System-level Losses, Hazards and Constraints.

| Unacceptable Losses | System-Level Threats/Hazards | System-Level Constraints |
|---|---|---|
| L-1: Death, dismemberment or injury to personnel | T-1: Attacker operates chiller plant beyond normal operational limits (temperature, pressure, flow-rate) <br> H-1: System is operated beyond normal operational limits (temperature, pressure, flow-rate) [L-1, L-2, L-3] | SC-1: System must not operate beyond normal operational limits |
| L-2: Physical damage to critical equipment | T-2: Attacker prevents chiller plant from meeting cooling load <br> H-2: System does not meet cooling load [L-3] <br> H-2.1: Inadequate throttling of cooling capacity <br> H-2.2: Inadequate distribution of chilled water to facility buildings <br> H-2.3: Inadequate rejection of waste heat to atmosphere | SC-2: System must be adequately controlled to meet cooling load |
| L-3: Loss of mission, i.e., inability to deliver chilled water | T-3: Attacker forces Chiller plant to violate order and/or timing sequence (permissive functions, lube oil system) <br> H-3: System violates order and/or timing sequence of operations [L-1, L-2, L-3] | SC-3: System must not violate order or timing sequence of operations |
| | T-4: Attacker causes a refrigerant leakage <br> H-4: System is unable to prevent contamination of environment with refrigerant leakage [L-1] | SC-4: System must prevent environment contamination due to refrigerant leakage |

## 4.2. Model the Functional Control Structure

One of the key aspects of the Cybersafety method is understanding the control structure that enforces constraints on the system to prevent it from moving into unsafe/insecure states. Figure 3 represents the functional control structure for the chiller plant; here a hierarchy of controllers are shown where each controller has a function along with certain safety and security-related responsibilities. The controllers fulfill their responsibilities by taking control actions based on some feedback with the goal of keeping the chiller plant safe.

The functional control of the chiller plant consists of not only managing the combined cooling capacity of all the chillers, but also the auxiliary equipment to enable distribution of chilled water to facility buildings as well as ejection of waste heat to the environment via cooling towers. While individual control of chiller compressors, chilled water pump motors and valves, cooling tower fans, etc. is implemented via Programmable Logic Controllers (PLC), increasingly employing Industrial Internet-of-Things (IIoT) technologies, the overall control logic for system operation is managed by the plant Distributed Control System (DCS). The DCS, through a Human–machine Interface (HMI), provides the plant operator with a birds-eye view of all the equipment in the plant and enables supervisory control of field equipment by transferring settings, operator permissive functions and manual override commands to field controllers.

Note that the plant's mandate is limited to maintaining chilled water supply at a certain temperature, pressure and flowrate; the control of Building Automation Systems (BAS) is beyond its mandate and is in fact controlled by a different group of operators, referred to as Facilities Operators (Figure 3). The Plant Operator actions, in turn, are controlled via operating procedures and instructions by Plant Engineers. Both Plant Engineers and operators report to plant's Operations Management which enforces the leadership's enterprise-level goals and vision through policies and standards. The leadership team, in turn, is controlled by municipal, state and federal regulations enforced via certificates and licensure for operating the plant.

The chiller plant does not operate in isolation; in fact, it is closely coupled with other systems in the plant, notably the electric generation and boiler systems. The operation of the chiller plant is contingent on decisions such as what combination of chillers should be run to achieve the desired cooling capacity, what is the desired chilled water setpoint and flow-rate, how many pumps should be operated and at what capacity, and which cooling towers should be operated and at what capacity. These decisions are highly dependent on environmental factors such as weather conditions, energy costs for electricity, gas, steam (imported from neighboring power plant) as well as cooling load (dependent on time of day and building occupancy).

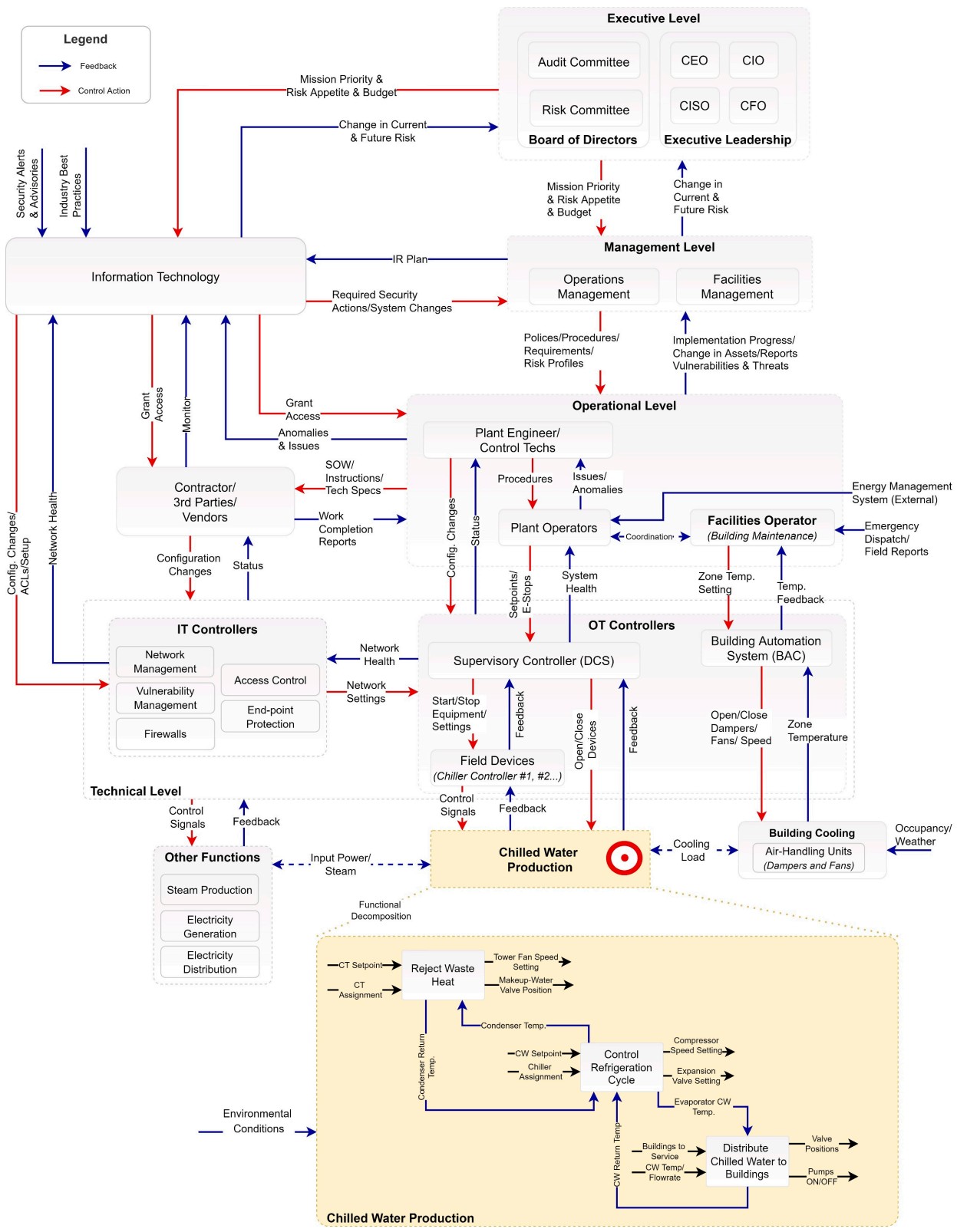

**Figure 3.** Detailed Hierarchical Functional Control Structure for the Chiller Plant.

Although each of the field devices are operated as individual components, they together interact in indirect ways to produce complex, emergent behavior that is greater than the sum of the parts. Each control decision is made in such a way so as to achieve a global optimum for the plant to maximize efficiency. An Energy Management System

(EMS) service provider is contracted to provide recommendations for optimum integrated performance. The EMS combines aggregated data from the plant's DCS, IIoT sensors and real-time market, weather and fuel prices with predictive analytics to recommend operating points that maximize efficiency for the plant. The EMS has the ability to automatically update the plant control parameters remotely as well; although this feature is currently not employed.

In addition to the operational technology (OT) field equipment (including PLCs, chiller controllers, HMI, etc.), there are a number of information technology (IT) controllers as well that are primarily responsible for enforcing security constraints on the system. These include IT tools such as network firewalls, identity management and access control, vulnerability and patch management software, network management, etc. The IT tools are primarily managed by the corporate IT team which is responsible for formulating and enforcing the IT security policy on the plant personnel and equipment.

For the purpose of this paper, we limit our focus to the chiller cooling capacity control loop. The chillers at the plant are equipped with a controller that regulates the cooling capacity of the chiller in response to chilled water temperature deviation from a set-point by adjusting the speed of the compressor motor [24]. The PLC receives feedback from several sensors, monitoring various physical processes, including refrigerant discharge and suction temperatures and pressures, condenser and evaporator water temperatures, pressures and flows, compressor lube oil temperature and pressure, guide vane position, etc., and computes the required compressor speed which is then implemented via a Variable Frequency Drive (VFD).

*4.3. Identify Unsafe/Insecure Control Actions*

The next step in the Cybersafety method is to identify Unsafe/Insecure Control Actions. We begin by identifying the primary functions, responsibilities and associated control actions for the main controllers in the functional control structure as presented in Table 2. Note that for each controller, we identify not only the safety-related responsibilities but also the security-related responsibilities and use the generic ICS architecture proposed by Open Secure Architecture (OSA) as a guide for this purpose [25].

**Table 2.** Partial List of Controllers, Safety Responsibilities and Control Actions (items in red are security related).

| Controller | Function Performed | Safety and Security Constraints | Control Actions |
|---|---|---|---|
| Chiller Controller | Control cooling capacity to achieve desired chilled water temperature setpoint and control auxiliary equipment | • Must ensure safety of *chiller* operation (permissive functions, sequencing) <br> • Must safely shutdown chiller during emergencies <br> • Must meet cooling load by adequately controlling chiller operation <br> • **Must prevent unauthorized changes to control parameters/logic** <br> • **Must maintain change log and record any changes to baseline configuration** <br> • **Must not establish connection with unauthenticated devices** | • Adjust compressor speed, metering device, guide vanes <br> • Operate aux equipment (valves, lube pump, cooling) <br> • **Restrict access to authorized personnel and devices only (user and device identification and authentication)** <br> • **Record any changes to baseline configuration** |

**Table 2.** *Cont.*

| Controller | Function Performed | Safety and <span style="color:red">Security</span> Constraints | Control Actions |
|---|---|---|---|
| Distributed Control System (DCS) | Manage chilled water supply by ramping up/down chillers to desired setpoints and operate auxiliary equipment | • Must ensure safety of *system* operation (by operating various field devices in correct order/sequence) <br> • Must alert operator if unsafe conditions exist <br> • **Must prevent unauthorized changes to control parameters/logic** <br> • **Must ensure integrity of control settings transferred to field devices** <br> • **Must not establish connection with unauthenticated devices** <br> • **Must provide only essential capabilities and restrict use of prohibited functions, ports, protocols and services** | • Adjust field device settings <br> • Raise faults and alarms <br> • **Restrict access to authorized personnel only; uniquely identify and authenticate users** <br> • **Prevent execution of unsafe commands** <br> • **Ensure transmission integrity during changes to field device settings** |
| Plant Operator | Perform day-to-day tasks to operate chiller plant along with other equipment including the gas turbine, boilers and auxiliary equipment to meet real-time demand variations | • Must ensure safe operation of plant equipment <br> • Must respond to alarms/faults/incidents to prevent damage <br> • **Must detect and escalate anomalous behavior to engineering** <br> • **Must respond to safety/security incidents to prevent damage** | • Manual overrides to keep equipment within safe limits <br> • **Respond to incidents/safely shut down equipment during emergencies** <br> • **Report anomalous behavior** |
| Plant Engineer | Act as the technical lead for plant operations; provide operating procedures, ensure compliance with procedures; provide tech specs to contractors/3rd parties; respond to incidents | • Must certify design, equipment and procedures for safe operation <br> • Must ensure procedural compliance and training <br> • **Must respond to safety and security incidents (preparation, detection and analysis of security issues, containment, eradication, and recovery)** <br> • **Must report security issues, anomalies to management** <br> • **Must prevent leakage of confidential designs, tech specs** <br> • **Must enforce change control procedures; generate, retain and review records reflecting any changes; maintain baseline config.** <br> • **Must ensure contractors have created a satisfactory security program** | • Equipment and design certification <br> • Approve operating procedures <br> • Provide technical specifications and requirements to contractors/vendors <br> • **Change control logic/config. of field devices** <br> • **Restrict access to sensitive information to authorized personnel only** <br> • **Ensure transmission integrity for changes to control logic/configuration** |

**Table 2.** *Cont.*

| Controller | Function Performed | Safety and Security Constraints | Control Actions |
|---|---|---|---|
| Information Technology (IT) Security | Ensure the plant's IT systems are free from malicious code, vulnerable systems are identified, reported and adequately mitigated | • **Must monitor network and endpoints for vulnerabilities and maintain effective vulnerability and patch management processes**<br>• **Must maintain up-to-date system inventory, configuration baselines and backups**<br>• **Must identify and prioritize security gaps and take remedial actions**<br>• **Must alert management and staff about threats and vulnerabilities**<br>• **Must restrict access to authorized personnel**<br>• **Must develop incident response plans**<br>• **Must prevent unauthorized changes to control parameters/logic** | • **Vulnerability Scanning**<br>• **Incident response assistance**<br>• **Access Control**<br>• **Patch Management**<br>• **Audits** |
| Contractors/ Vendors/ 3rd Parties | Provide equipment, services, software and maintenance for safe operation of the plant | • Must comply with plant policies and procedures<br>• **Must implement security processes to identify and manage system vulnerabilities and ensure secure deployment of products and services at client site** | • Issue config. changes at client site<br>• **Develop system security program**<br>• **Monitor service/product deployment** |
| Plant Management | Ultimately responsible for safe and secure operation of the plant | • Must ensure plant complies with all safety/security regulations<br>• Must allocate adequate resources and training for safe and secure plant operation consistent with risk profile<br>• **Must enforce physical and logical access restrictions**<br>• **Must enforce change control, incident response (IR) policies**<br>• **Must perform security risk assessments and manage risk**<br>• **Must provide effective oversight of plant security operations** | • Allocate budget<br>• Set priorities<br>• **Audit plant processes and procedures**<br>• **Provide training (Change control/IR)** |

Next for each controller, we identify unsafe and insecure control actions. The unsafe control actions can be thought of as a refinement of the system-level hazards/threats identified earlier. Note that a particular control action in of itself may not be unsafe, but the context in which it is performed, makes it safe or unsafe. Some unsafe and insecure control actions for the chiller plant are listed in Table 3. Technical details for some of these unsafe control actions are explained next.

**Table 3.** Partial List of Unsafe and Insecure Control Actions for some controllers (Insecure Control Actions are listed in red).

| Action By | Control Action | Not Providing Causes Hazard | Providing Causes Hazard | Too Soon, Too Late, Out of Order | Stopped too Soon, Applied too Long |
|---|---|---|---|---|---|
| Chiller Controller | Compressor Speed | UCA-1: Chiller controller does not reduce speed when compressor motor is overheated → [H-1.2] | UCA-3: Chiller controller increases speed when the required lift (pressure differential) is too high (suction pressure too low or discharge pressure too high) causing surging → [H-1.2] | UCA-6: Chiller controller increases speed before lube oil permissive function is available (lube oil not at correct temp/pressure) → [H-2, H-3] | UCA-9: Chiller controller continues to increases speed when refrigerant superheat is too low (liquid refrigerant is drawn into compressor) → [H-1] |
| | | UCA-2: Chiller controller does not increase speed when chilled water temp is below setpoint (unable to satisfy cooling load) → [H-1.3] | UCA-4: Chiller controller operates compressor in reverse direction → [H-1] UCA-5: Chiller controller toggles compressor speed between upper and lower limits repeatedly causing it to pass through resonant speeds→ [H-1] | UCA-7: Chiller controller increases speed before evaporator/condenser flow is established → [H-2] UCA-8: Chiller controller increases speed when timer permissive function is unavailable (causing damage from inrush current) → [H-3] | UCA-10: Chiller controller increases speed for too long after discharge pressure is beyond high-pressure cut-out → [H-1.2] |
| | <span style="color:red">Security Controls (Access Control/ Baseline Config.)</span> | <span style="color:red">UCA-1S: Chiller controller does not restrict unauthorized access to prevent modification of control settings → [H-1] UCA-4S: Chiller controller does not have a baseline configuration, i.e., does not record which version of program is running → [H-1]</span> | <span style="color:red">UCA-2S: Chiller controller connects with any device on the industrial network → [H-1]</span> | | <span style="color:red">UCA-3S: Chiller controller allows wider access to control settings than is required for regular operation → [H-1]</span> |
| Supervisory Controller (DCS) | Setpoints/Sequence of Operation | UCA-1: DCS does not start additional chiller(s) when cooling capacity has been reached (unable to satisfy cooling load) → [H-1.3] UCA-7: DCS does not open make up water valve when water level in the cooling tower is too low → [H-1] | UCA-2: DCS provides incorrect chilled water setpoint to chiller controllers → [H-1] UCA-4: DCS starts chilled water pump when the valve is closed (cavitation)→ [H-1] UCA-6: DCS operates chilled water pumps at resonant frequency → [H-1] | UCA-3: DCS energizes chiller starter motor bypassing the chiller controller (accidental start-up of chiller motor can cause severe damage to chiller) → [H-2] UCA-5: DCS opens isolating valve of additional chiller too quickly, effectively reducing flow-rate of incumbent chiller by half (potentially causing freezing) → [H-1.1] *refer to pp. 76 of [23] for additional details | |
| | <span style="color:red">Security Controls (Identity and Auth/Least Functionality/Privilege)</span> | <span style="color:red">UCA-1S: DCS does not prevent unauthorized access to modify supervisory control settings → [H-1] UCA-4S: DCS does not prevent installation/ execution of malicious code → [H-1]</span> | <span style="color:red">UCA-2S: DCS allows configuration of ports, protocols, and/or services that are beyond organization-defined list of approved services → [H-1]</span> | | <span style="color:red">UCA-3S: Chiller controller allows wider access than is required for regular operation → [H-1]</span> |

**Table 3.** *Cont.*

| Action By | Control Action | Not Providing Causes Hazard | Providing Causes Hazard | Too Soon, Too Late, Out of Order | Stopped too Soon, Applied too Long |
|---|---|---|---|---|---|
| Plant Operator | Start/Stop Equipment | UCA-1: Operator does not shut down equipment when hazardous conditions occur → [H-1] | UCA-2: Operator bypasses safeties by forcing flags/overriding permissive functions → [H-1] | UCA-3: Operator provides hazardous manual inputs via HMI/DCS out-of-order → [H-2] | UCA-4: Operator changes chiller controller mode to manual and leaves it in manual (for too long) when controller feedback is not following correct sequence → [H-2] |
| | Security Controls | UCA-1S: Operator does not report anomalous behavior to IT/Engineering for investigation → [H-1] UCA-3S: Operator does not adequately respond to a security event losing critical forensics information preventing further investigation → [H-1] | UCA-2S: Operator downloads malicious files to supervisory control system → [H-1] | | |
| Plant Management | Issue Policies/Funds | UCA-1S: Management does not enforce an effective path/vulnerability management program → [H-1] UCA-5S: Management does not have an adequate risk assessment/management program for plant security UCA-7S: Management does not provide effective security oversight; does not maintain an effective incident response plan | UCA-2S: Management approves unsafe/insecure procedures to keep production running/during emergencies/contingencies → [H-1, H-2, H-3] UCA-6S: Management prioritizes ineffective security controls → [H-1, H-2, H-3] | UCA-3S: Management disburses funding for training, mitigation strategy implementation too late → [H-1] | UCA-4S: Management does not follow its incident response plan → [H-1] |

- Pump/Compressor Critical Speed and Reverse Rotation

The compressor motor has certain critical speeds at which mechanical resonance can occur. Typically, the VFD is programmed to skip over these resonant frequencies [26]. However, operating the motor at its critical speed, can cause considerable damage to 'the bearings and the motor shaft' [27] [UCA-5]. Another unsafe condition for the compressor motor is reverse rotation; the VFD can be easily toggled to change the direction of rotation. Although reversing the direction of rotation would not change the direction of fluid flow through the compressor, it would cause significant damage to the compressor due to vibrations [UCA-4, UCA-5].

- Lubrication Oil

A Centrifugal compressor needs oil forced around its internal components (such as gears, thrust bearings, etc.) to provide lubrication and remove heat caused by friction. The lubrication oil has to be at the correct temperature and pressure for it to perform its intended function; it must be thin enough to lubricate properly at high speeds of rotation but also thick enough to handle the heat and refrigerant contamination that can occur. If the lubrication oil conditions are not at the correct temperature and pressure, it can destroy the compressor in a matter of a few minutes because of the excessive build-up of heat through friction in the internal components [28] [UCA-6].

- Motor Burnout

If a compressor motor is operating at its temperature or current limit, a command to increase motor speed would result in overheating; excessive heat can lead to premature loss of motor winding insulation, resulting in the motor burning itself out [UCA-1, UCA-8].

- Surging

Another characteristic hazardous condition for centrifugal chillers is surging. This can occur when the compressor differential pressure exceeds design limits, particularly during low-load operation; it is caused when the required lift exceeds the systems pumping capacity. It may be caused by either increasing the condenser temperature and pressure or reducing the evaporator temperature and pressure—both could be caused by reducing water flowrate in the condenser or evaporator at low load conditions.

Once surging occurs, the output pressure of the compressor is drastically reduced, resulting in flow reversal within the compressor. The flow reversal applies significant dynamic forces on the impeller which subjects the compressor components (such as thrust bearings, bearings, casing) to large axial force changes due to the rotor rocking back and forth. If not controlled it can cause tight-tolerance compressor internals to be permanently damaged due to asymmetric thermal expansion and subsequent friction damage [29] [UCA-3].

While there are several other hazardous control actions, we selected a small subset to demonstrate the diversity of hazardous control actions that need to be considered in the analysis. Note that the unsafe/insecure control actions reported in Table 3 are not limited to the chiller cooling capacity control loop but cover other controllers (such as Plant Operator, Management, etc.) at various hierarchical levels of the system.

### 4.4. Generate Loss Scenarios

Next, we determine causal factors that enable the issuance of the earlier identified unsafe/insecure control actions; we want to understand 'why', in the context of the larger system, an unsafe/insecure control action may be issued by a controller. According to Leveson [14], two types of causal scenarios should be considered:

(a) Scenarios that lead to the issuance of unsafe control actions; these could be a result of (1) unsafe controller behavior or (2) inadequate/malformed feedback.
(b) Scenarios in which safe control actions are improperly executed or not executed altogether; these could be a result of issues along the (1) control path or the (1) controlled process itself.

For illustration purposes, we zoom into the functional control structure for the chiller controller from Figure 3 and superimpose it with guidewords from [30], signifying sample attack scenarios; the simplified control structure is presented in Figure 4. By going around the control loop and hypothesizing why a controller may issue a hazardous control action while considering the actions and motivations of malicious actors, we can generate a list of causal factors for loss scenarios. Two example loss scenarios, UCA-5 and UCA-3S, along with potential causal factors and associated safety/security constraints are presented in Table 4 which is followed by a discussion of some of the key findings.

**Table 4.** Partial List of Loss Scenarios.

| Chiller | | UCA-5 |
|---|---|---|
| Chiller controller toggles compressor speed between upper and lower limits repeatedly causing it to pass through resonant speeds → [H-5]—(Cooling Capacity Control Loop) | | |
| **Scenarios** | **Associated Causal Factors** | **Safety/Security Constraints** |
| 1 — Tampered or Fabricated Sensor Signal: Chiller controller operates the compressor at unstable speeds (increasing and decreasing the speed repeatedly) as it passes through resonant frequencies because it is fed with tampered chilled water temperature feedback (man-in-the-middle (MITM) or stealthy manipulation of sensors), e.g., (a) Network-based malicious spoofing of the temperature sensor values transmitted to the controller, e.g., the attacker alters chiller controller's I/O register values that store the current temperature state. (b) Electro-magnetic interference injection attack on the thermocouple [31] | 1. Process/Mental Model Flaw <br> • Controller intrinsically trusts the sensor values—it believes the compressor needs to be sped up or slowed down to meet cooling load demand without verification <br> • Belief that the attack on the process can only occur via the network not remotely via EMI attack <br> 2. Contextual Factors <br> • Plant employs MODBUS protocol to transmit data between controllers and DCS (which can suffer MITM attacks) <br> • Poor access control—employees have unrestricted access to plant equipment <br> 3. Structural/Control Flaws <br> • No physical controls in place to prevent EMI attacks <br> • No intrusion detection and monitoring at Level 0 <br> 4. Dynamics and Migration to Higher Risk <br> • The facility does not employ out-of-band verification using power meters (i.e., measure compressor current draw that would independently indicate such anomalous behavior) | 1. Must employ out-of-band verification, e.g., independently monitor compressor current draw to detect anomalous behavior <br> 2. Must have physical shielding against EMI attacks for critical sensors and physical access control (access to control cabinets) must be bolstered <br> 3. Must employ Endpoint Detection and Response (EDR) solution |
| 2 — Unauthorized Changes to Control Algorithm: Chiller controller operates the compressor at unstable speeds (increasing and decreasing the speed repeatedly) as it passes through resonant frequencies because of unauthorized changes to its control algorithm (i.e., incorrect Proportional, Integral and Derivative (PID) values leading to controller instability). This could be caused by an upload of malicious code by: (a) Malicious Insider (b) Cyber-attacker (c) Contractor/3rd Party (see Scenario #3) | 1. Process/Mental Model Flaw <br> • Belief that the control logic and all parameter values are legitimate—no scheduled verification of code <br> 2. Contextual Factors <br> • Engineering workstation has access to source code for all programs running at site (more than what is needed) <br> • PLCs are connected to engineering workstation to enable diagnostics and programming <br> 3. Structural/Control Flaws <br> • Inadequate control on PLC program change management (i.e., anyone with access to engineering workstation can upload a program to the PLC) <br> 4. Dynamics and Migration to Higher Risk <br> • PLCs have physical security features to prevent remote programming. However, to facilitate convenience, the sites frequently enable the remote programming feature on the PLCs | 1. Must employ digital signatures or Checksums to ensure integrity of code running on field devices at a regular frequency <br> 2. Must ensure that access to programs running on site is severely restricted and follows the need-to-know information security principle; must not store all PLC programs in one place (i.e., the Engineering Workstation) <br> 3. Disable remote programming and web access/email functionality in PLCs |
| 3 — Unauthorized changes to Control Algorithm—Contractor: During routine maintenance (firmware update), contractor inadvertently uploads malware that changes the chiller controller's control algorithm | 1. Process Model Flaws <br> • Belief that contractor has been adequately vetted and follows the same rigorous security procedures as the asset owner <br> 2. Contextual Factors <br> • Contractor/vendor uses removable media (USBs, Laptop) without scanning for malware, thereby inadvertently uploading malicious firmware during routine maintenance <br> 3. Structural/Control Flaws <br> • Vendor activities are not closely monitored or audited (because of lack of understanding of vendor/contractor activities) | 1. Must demonstrate customer adherence to company's cybersecurity policy prior to connecting to plant equipment <br> 2. Must strictly limit access to required assets and resources to execute the task <br> 3. Must implement unidirectional data transfer diode which leverages hardware features to restrict bi-directional data transfer |

**Table 4.** *Cont.*

| Chiller | | | UCA-5 | | |
|---|---|---|---|---|---|
| 4 | Incorrect control input from a higher-level controller: Chiller controller operates the compressor at unstable speeds (increasing and decreasing the speed repeatedly) because of an incorrect input from a higher-level controller. The chiller HMI is under control of an attacker who sends malformed commands via a MITM attack | 1.<br><br>2.<br><br>3. | Process/Mental Model Flaw<br>• Belief that a command originating from the HMI is always legitimate and must be executed<br>Contextual Factors<br>• MODBUS communication protocol is used between the chiller controller and the HMI—there is no authentication or encryption; commands can be hijacked by an attacker<br>Structural/Control Flaws<br>• No hard-coded limits on the number of times the chilled water setpoint can be safely changed within a given amount of time | 1.<br><br>2. | Must implement hard coded limits on the number of times the compressor can be ramped/unloaded<br>Must employ proper network segmentation to prevent HMI access from the business network or the internet |
| 5 | Malformed actuator implementation: Critical resonant frequencies on the Variable Frequency Drive (VFD) are improperly set; toggling the compressor between upper and lower limits, causes the compressor to pass through critical speeds resulting in compressor damage. This could be caused by:<br>(a) Attacker resets the VFD critical speed settings<br>(b) Plant personnel not setting up the VFD properly at commissioning or insider attack | 1.<br><br>2. | Contextual Factors<br>• VFD employed at site allows read/write functionality of critical speeds over the network; attacker reprograms VFD with incorrect critical speeds that enables the controller to operate at resonant speeds<br>Structural/Control Flaws<br>• Commissioning after maintenance does not involve verification of critical frequencies due to lack of inhouse capability (training) and assumption (mental model flaw) that critical speeds remain unchanged<br>• No policy/engineering specification exists against the purchase/use of VFDs with network functionality | 1.<br><br>2.<br><br>3.<br><br>4. | Must not allow VFD to have read/write functionality over internet<br>Must disable default web access to VFD (including ability to send emails over insecure ports)<br>Management policy should prevent purchase of VFDs with network functionality<br>Must validate VFD critical speed setting on all chillers after every outage/refurbishment/ maintenance cycle |

| Operator | | | UCA-3S | | |
|---|---|---|---|---|---|
| | | Operator downloads malicious files to supervisory control system → [H-1] | | | |
| | Scenarios | | Associated Causal Factors | | Safety/Security Constraints |
| 6 | Incomplete/Tampered Process Model: Operator downloads a malicious file to the supervisory control system due to an incomplete/tampered process model | 1.<br><br>2.<br><br>3. | Process/Mental Model Flaw<br>• Flawed belief that the file is safe to download; operator does not suspect the file to be malicious (poor training) or operator is led to believe the file is legitimate (no feedback/alert to raise suspicion)<br>• Belief that the plant is air-gapped so copying/downloading unverified files to DCS is safe<br>Coordination and Communication Issues<br>• Lack of effective communication between IT and OT results in misunderstanding of security controls<br>Contextual/Environmental Factors<br>• Less oversight on night shifts along with less stringent work—operator more susceptible to downloading files for entertainment (music, videos) | 1.<br><br>2.<br><br>3. | Must ensure operator training and awareness about cyber risks<br>Must ensure effective communication between IT/OT to improve understanding of cross-domain risks<br>Must prevent downloading of files onto control systems (e.g., by physically blocking open ports) |
| 7 | Inappropriate, ineffective or missing control actions:<br>Operator downloads malicious files to DCS because of ineffective or missing control actions | 1.<br><br>2. | Process/Mental Model Flaw<br>• Belief that malicious files would be detected and blocked with IT controls but IT does not regularly patch OT equipment<br>Structural/Control Flaws<br>• No controls to prevent downloading files to DCS<br>• No controls to verify baseline configuration of DCS/SCADA system | 1. | Must regularly verify baseline configuration of DCS/SCADA system and other assets to detect unauthorized changes |

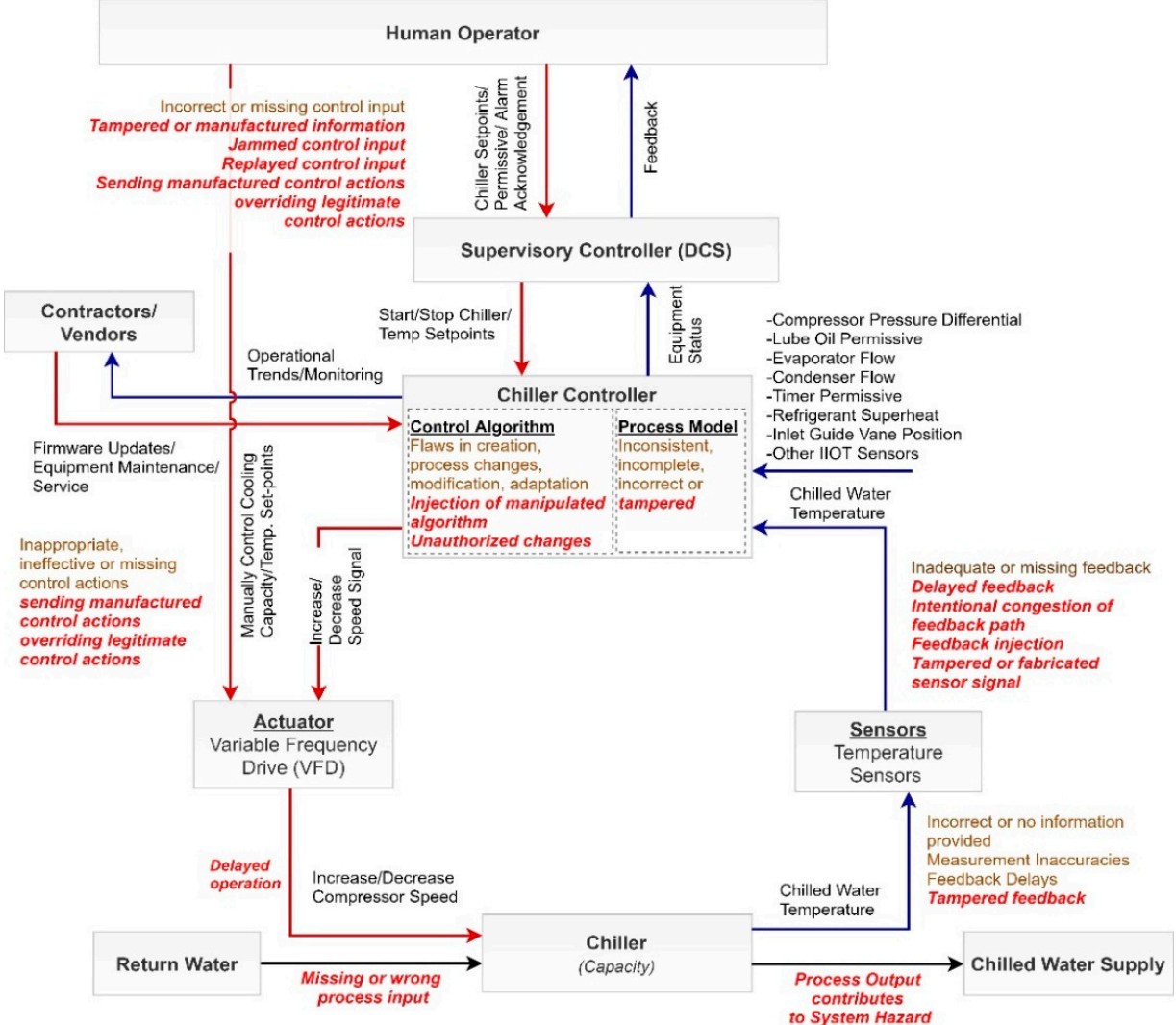

**Figure 4.** Simplified Control Structure for Chiller Control Loop with Sample Attack Scenarios.

## 5. Discussion

In the previous section, we generated detailed scenarios for two unsafe/insecure control actions and identified a number of causal factors. Note that despite the limited scope of these scenarios, they provide a number of interesting insights about the chiller plant that go beyond technical vulnerabilities and span the larger socio-organizational system. Not only that, but this method allows a deeper introspection about the reason for the existence of the vulnerabilities, including flaws in the control structure as well as process/mental models flaws. Based on this deeper understanding of the vulnerabilities, new safety and security constraints are defined not only for individual components but also for their interactions, ultimately making the chiller plant more resilient to cyberattacks.

As shown in Table 4, the unsafe control action of toggling compressor speed between upper and lower limits, can be caused by a number of scenarios. For instance, Scenario #1 indicates how the feedback from the process sensors could potentially be tampered by an adversary to ramp the compressor speed up or down. Such a scenario could occur via an electromagnetic injection attack (described in detail by Yu et al. [31]) or by manipulating the Modbus I/O registers on the controller. To engineer out this weakness, the facility must employ a combination of technical and process changes. For instance, it should explore employing out-of-band verification for critical process sensor values (e.g., use power meters to monitor compressor current draw) along with deploying intrusion

detection and monitoring solution at field device level or ensuring analog sensors are adequately shielded.

Scenario #2 describes how unauthorized changes to the chiller controller could cause compressor instability. This scenario explores how inadequate controls for accessing PLC programs and lack of programmatic verification of PLC programs/parameters and version control could be exploited by an adversary. This scenario is possible because of poor change management practices and poor cyber hygiene. To preclude the possibility of this scenario, the facility must severely restrict access to PLC programs, disable all web access functionality to PLCs and employ a process to regularly verify PLC programs running on site. Scenario #3 is related to Scenario #2, except that the unauthorized changes are caused by 3rd Party/Contractor/Vendor. This scenario explores how lack of understanding of contractor's scope of work coupled with poor enforcement of asset owner's security policies can result in vulnerabilities for critical assets. This scenario can be mitigated by strictly enforcing security policies on contractors, vendors and 3rd parties and limiting access to equipment and resources that are essential to complete the job.

Scenario #4 is similar to Scenario #1, except that instead of tampering sensor feedback, the attacker directly targets the chiller remote HMI to alter compressor speed. This scenario could be mitigated by enforcing hard-coded limits on the number of times the chiller compressor speed can be adjusted in a given time duration. This would preclude both insider threats as well as external threats. If this engineering solution is coupled with proper network segmentation (that prevents the attacker from reaching the HMI), it would significantly reduce the likelihood of such a scenario.

Scenario #5 underpins each of the other scenarios; it describes how the use of a Variable Frequency Drive (VFD) with remote programming functionality can be exploited by an adversary to reprogram the critical resonant frequencies. This vulnerability exists because of a lack of management policy restricting the use of a VFD with such functionality. Investigating further, the lack of management policy could be a result of a flawed process model which would in turn be a result of an inadequate risk assessment feedback from external cybersecurity consultants. This weakness could be mitigated by barring the use of VFDs with such functionality through management policy or at the very least disabling network access feature.

Scenarios #6 and #7 describe an insecure control action undertaken by the operator, i.e., downloading a malicious file onto the control system. Several interesting causal factors are identified, e.g., the flawed belief that the file is not malicious or the flawed belief that a malicious file would be prevented from downloading by IT controls. Some interesting contextual factors are also highlighted; for instance, operators on night shifts tend to have less oversight and hence more likely to download malicious files to their computers. In one case study, we discovered that night-shift operators had installed movies, music and games from suspicious sites on the SCADA system (to ward off boredom during downtime). We also identify control flaws that would enable this insecure action, i.e., no controls against downloading files and no programmatic verification of SCADA/DCS baseline configurations. The point is that with this deeper understanding about the system, we can design the system to mitigate such vulnerabilities emerging from user actions.

So far, we explored loss scenarios for two unsafe/insecure control actions. The same approach would be repeated for each of the other controllers. Figure 5 provides a summary of safety/security constraints, control flaws, process/mental model flaws for each of the controllers. This provides a holistic, systems-view of the various flaws in assumptions and controls at the chiller plant that can ultimately lead to security incidents.

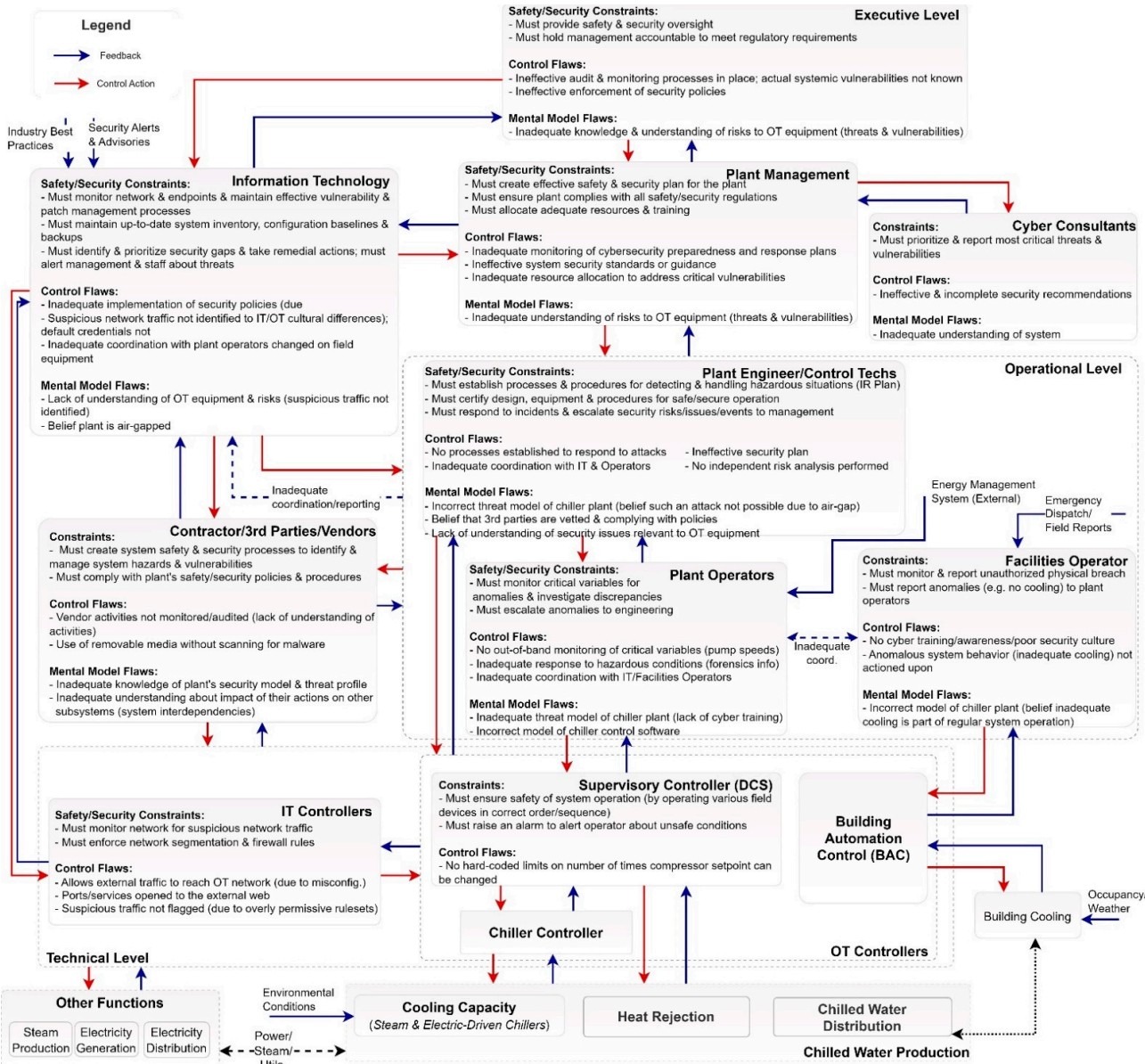

**Figure 5.** Summary of flaws in the larger socio-organizational structure.

## 6. Final Remarks

With the advent of advanced cyber adversaries and nation-state attackers, critical infrastructure industrial control systems are under threat. We urgently need to rethink, re-engineer and redesign our industrial control systems using a holistic, systematic approach in order to ensure system resilience and operation under adversity. While the traditional approach to protect such critical systems is to take a probabilistic risk-based approach, focused on technical aspects of the system, in this paper, we demonstrated how a system-theoretic approach could be applied to an archetypal industrial control system (i.e., industrial chillers) to elicit safety and security requirements. To the best of our knowledge this is the first integrated safety and security analysis of industrial chillers using the system-theoretic approach.

In addition, we identified a gap in the current application of STAMP-based system-theoretic methods (*STPA-Sec/Cybersafety*, etc.) in that they were essentially safety analyses that identified cybersecurity causal factors. We believe security has to be considered as an integral part of the safety control structure with explicit identification of security-related roles and responsibilities, constraints, control actions and loss scenarios. To that end, the

proposed improvements to the STAMP-based *Cybersafety* method (by focusing on threats, security constraints and controllers), provides a well-guided and structured approach to develop an integrated safety and security model to holistically identify cyber-vulnerabilities and mitigation requirements in complex industrial control systems.

Importantly, as shown in this paper the scope of the identified vulnerabilities and mitigation requirements span not only technical aspects of the system but also weaknesses that emerge from interactions in the larger socio-organizational system. These could be considered examples of indirect interactions between components of the system that create conditions necessary for a security/safety event. It is worth noting that one attack scenario almost identical to the one described in this paper occurred recently when a 300-ton chiller was destroyed by loading/unloading the chilled water pumps in a commercial building [32], lending credence to the significance/applicability of this work to real-world use-cases. Finally, we recognize, that despite its advantages, the Cybersafety method does not provide any indication of prioritization of loss scenarios unlike risk-based methods that provide a risk score. In future work, we will explore how the *Cybersafety* method can be improved to prioritize loss scenarios so that security leaders can address the most critical weaknesses in their systems in order of prioritization.

**Author Contributions:** Conceptualisation, S.K. and S.M.; investigation, S.K. and S.M.; methodology, S.K and S.M.; supervision, S.M.; writing—original draft, S.K.; writing—review and editing, S.K. and S.M. All authors have read and agreed to the published version of the manuscript.

**Funding:** This material is based upon research supported by the Department of Energy under Award Number DE-OE0000780, along with funds from "Fondo Europeo di Sviluppo Regionale Puglia POR Puglia 2014–2020—Asse I–Obiettivo specifico 1a—Azione 1.1 (R&S)—Titolo Progetto: Suite prodotti CyberSecurity e SOC" and BV TECH S.p.A., a seed grant from the MIT Energy Initiative (MITei), and funds from the corporate members of Cybersecurity at MIT Sloan: The Interdisciplinary Consortium for Improving Critical Infrastructure Cybersecurity.

**Conflicts of Interest:** The authors declare no conflict of interest.

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
