# Peer review of "Protecting Chiller Systems from Cyberattack Using a Systems Thinking Approach"

_2673-8732, doi:10.3390/network2040035_

Round 1

Reviewer 1 Report

The authors presented a scenario in which they applied the several steps of the Cybersafety framework to protect Chiller Systems. In fact, security in systems is a very interesting topic, but the authors should improve some aspects in the article.

--

In Chapter 1 "Introduction", the authors provide an introduction and background to the topic.

The authors should make clear what the scientific contribution and objectives of this work are. 

At the end of this chapter, the authors should add a paragraph explaining how the paper is organized.

--

Section 1.1 "Literature Review", should be improved. The authors should elaborate a comparison of the works related to the proposed solution. They should highlight the methods used, strengths and weaknesses of each solution. 

After this improvement, the possibility of the Literature Review being chapter 2 should be considered.

--

Cybersafety Method (Section 1.2) was it the method defined and created by the authors?

If yes, it should have more focus and should be presented in Chapter 3 ( Considering that CAP 2 was reserved for the analysis of related works). 

--

Chapter 2 "Description of the System" should be a subsection of Chapter 3. So the chapter starts with a description of the system that will later be targeted for application of the cybersafety method.

--

Authors should pay attention to the size of figures and tables so that they do not exceed the document margins.

--

Finally, Chapter 5 "Conclusions", like the introduction, should be improved. The authors should clarify the scientific contributions of the paper.

--

Author Response

Response Letter

For Network-1891085:

“Protecting Chiller Systems from Cyberattack using a Systems Thinking Approach”

Thank you for reviewing the paper and providing very valuable comments and encouragement. Based on the comments and suggestions, we have revised and improved the manuscript. We highlight the major revisions in the main manuscript. We hope that you would find the current revision acceptable.

Importantly, as noted by one of the reviewers the scientific contribution and objectives of the work were not clear. The reviewer had also suggested reorganizing the paper and improving the literature review. In this version of the manuscript, we have addressed all these concerns. Specifically, we have clarified that:

  • Provides an integrated safety and security model. Hitherto, security analyses based on STAMP framework have focused on safety analysis with cybersecurity causal factors.
  • This is (to the best of our knowledge) the only application of STAMP/STPA-Sec/Cybersafety on an industrial chiller plant which is an archetypal industrial control system.    

For the convenience of reviewers, we list each comment and detail our revision. In addition, we proof-read the entire paper to eliminate typographical errors.

Looking forward to hearing back from you. 

Yours,

Shaharyar Khan

On behalf of Stuart Madnick

---------------------

# Reviewer 1

-----------------

  • In Chapter 1 "Introduction", the authors provide an introduction and background to the topic. The authors should make clear what the scientific contribution and objectives of this work are. 

Response 1: In this version of the paper, we have clarified the scientific contribution of this work. Specifically, the primary contribution of this work is to present an integrated safety/security model for an industrial control system application. While others claim to perform security analyses based on the STAMP framework, we argue that those are essentially safety analyses that identify cybersecurity causal factors. In this work, we create an integrated model by including security-specific requirements at every level of the analysis. To elucidate this point further, consider the controls in place when a security incident happens. The traditional STAMP methods would look at the PLC, operator etc., and determine how each of these could be causal factors for the attack to succeed. We are saying that in addition to that, we also need to look at security controls in place (their roles, responsibilities and actions) in addition to safety controllers/decision-makers. In some instances, the traditional safety controllers could have security responsibilities as well. In our analysis we explicitly identify security responsibilities for each controller.  

  • At the end of this chapter, the authors should add a paragraph explaining how the paper is organized.

Response 2: We have implemented this change and believe this has improved this manuscript.

  • Section 1.1 "Literature Review", should be improved. The authors should elaborate a comparison of the works related to the proposed solution. They should highlight the methods used, strengths and weaknesses of each solution. After this improvement, the possibility of the Literature Review being chapter 2 should be considered.

Response 3: We have completely revised this section and presented a brief comparison of the various works/methodologies related to the proposed solution. We have also transformed the literature review section into a standalone section as advised.

  • Cybersafety Method (Section 1.2) was it the method defined and created by the authors? If yes, it should have more focus and should be presented in Chapter 3 ( Considering that CHAP 2 was reserved for the analysis of related works). 

Response 4: The Cybersafety method is based on the STAMP framework which is described in detail in Prof. Leveson’s work. We have included the following text in Chapter 3 to clarify this point and kept this Chapter relatively short to keep the focus on the differences between our proposed approach and the classical STAMP-based approaches:
“The STAMP-based STPA method is described in detail in the STPA handbook [14] while its application to a real-world industrial control system cybersecurity example is de-scribed as the Cybersafety method by Khan & Madnick [6]. In this section, we describe the proposed improvements to the Cybersafety method to better elicit security requirements from the analysis.”

We believe this addresses the reviewer’s comment.

  • Chapter 2 "Description of the System" should be a subsection of Chapter 3. So the chapter starts with a description of the system that will later be targeted for application of the cybersafety method.

Response 5: We have reorganized this Chapter 3 to incorporate this change.

  • Authors should pay attention to the size of figures and tables so that they do not exceed the document margins.

Response 6: We reformatted our figures and tables to ensure they do not exceed the document margins. 

  • Finally, Chapter 5 "Conclusions", like the introduction, should be improved. The authors should clarify the scientific contributions of the paper.

Response 7: We have completely rewritten the conclusion and incorporated this feedback. We have reworded this section as “Final Remarks”. 

Reviewer 2 Report

The paper presents what must be taken into account to create a safe installation (chiller plant) that has a robust functionality. In the current context of Industry 4.0 and 5.0, the article is up-to-date. I recommend to explain all acronyms used in paper  (e.g. "TJX", row 94) as a list of acronyms or when was used. The chiller plant taken as an example is well described and vulnerabilities are well defined. Figure 5 which describes the control process of the cooling  plant is exhaustive (even is about maintaining chilled water supply at a certain temperature, pressure and flowrate). Scenarious in Table 4 (row 398) must be sift according with specific case. For example , on first scenarious has "Electro-magnetic interference injection attack on the thermocouple". Thermocouples are built in a metal frame. It is more relevant if the connecting conductors are investigated (shielding or not, short or long length, accesible or not). 

Author Response

Response Letter

For Network-1891085:

“Protecting Chiller Systems from Cyberattack using a Systems Thinking Approach”

Thank you for reviewing the paper and providing very valuable comments and encouragement. Based on the comments and suggestions, we have revised and improved the manuscript. We highlight the major revisions in the main manuscript. We hope that you would find the current revision acceptable.

Importantly, as noted by one of the reviewers the scientific contribution and objectives of the work were not clear. The reviewer had also suggested reorganizing the paper and improving the literature review. In this version of the manuscript, we have addressed all these concerns. Specifically, we have clarified that:

  • Provides an integrated safety and security model. Hitherto, security analyses based on STAMP framework have focused on safety analysis with cybersecurity causal factors.
  • This is (to the best of our knowledge) the only application of STAMP/STPA-Sec/Cybersafety on an industrial chiller plant which is an archetypal industrial control system.    

For the convenience of reviewers, we list each comment and detail our revision. In addition, we proof-read the entire paper to eliminate typographical errors.

Looking forward to hearing back from you. 

Yours,

Shaharyar Khan

On behalf of Stuart Madnick

---------------------

# Reviewer 2

-----------------

  • In the current context of Industry 4.0 and 5.0, the article is up-to-date. I recommend to explain all acronyms used in paper (e.g. "TJX", row 94) as a list of acronyms or when was used.

Response 8: We scanned the manuscript to ensure it did not contain any acronyms that were not explained. We also included footnotes on page 4 to describe some of the acronyms used in the manuscript.

  • Scenarious in Table 4 (row 398) must be sift according with specific case. For example , on first scenarious has "Electro-magnetic interference injection attack on the thermocouple". Thermocouples are built in a metal frame. It is more relevant if the connecting conductors are investigated (shielding or not, short or long length, accesible or not).

Response 7: Thank you for the comment. Unfortunately, we did not fully understand this comment. Our intent in Table 4 is not to prescribe specific solutions or identify detailed attack paths. Rather our intent is to highlight how one can utilize the control loop diagram to identify scenarios at a high-level that must be investigated by security practitioners. However, in order to further clarify electromagnetic interference (EMI) injection attack scenario, we have included a reference to the work done by Y. Tu et. Al [1] who have explored this attack scenario in significant detail along with potential mitigation. We have also referenced this work in the discussion section.

[1] Y. Tu, A. Rodriguez, S. Rampazzi, K. Fu, B. Hao, and X. Hei, “Trick or heat? Manipulating critical temperature-based control systems using rectification attacks,” Proc. ACM Conf. Comput. Commun. Secur., pp. 2301–2315, Nov. 2019.

Reviewer 3 Report

he authors have based this work on their 2021 paper [5] where they introduce a term cybersafety based on the STAMP (System-Theoretic Accident Model and Processes) framework. The main contribution is the application to an industrial chiller plant.

The main difference between STAMP and other risk frameworks appears to be incorporation of feedback based control, which makes the approach suitable for an industrial control systems (ICS).

They have proposed additions to the cybersafety method  to address security issues and uncover systemic vulnerabilities. The additions are given in red, which makes them clearly visible.

They start by compiling lists of System-level Losses, Hazards and Constraints (Table 1),, which is followed by detailed model of the control structure of the Chiller (Fig 3). They then identify unsafe/insecure control actions. They then consider sample attack and loss scenarios (Fig 4, Table 4). They identify the flaws in the socio-organizational structure.

There exist a number of risk frameworks intended for computations of risks, including cyber-security risks. This paper does not address quantitative evaluation however it attempts to identify potential system vulnerabilities. 

This paper is based on their earlier work which does not seem to have drawn significant attention yet. 

How does this approach compare with other related approaches?

The paper appears to be generally well written. There are some occasional typos (for example line 105: "safety control structure for the syste").

Author Response

Response Letter

For Network-1891085:

“Protecting Chiller Systems from Cyberattack using a Systems Thinking Approach”

Thank you for reviewing the paper and providing very valuable comments and encouragement. Based on the comments and suggestions, we have revised and improved the manuscript. We highlight the major revisions in the main manuscript. We hope that you would find the current revision acceptable.

Importantly, as noted by one of the reviewers the scientific contribution and objectives of the work were not clear. The reviewer had also suggested reorganizing the paper and improving the literature review. In this version of the manuscript, we have addressed all these concerns. Specifically, we have clarified that:

  • Provides an integrated safety and security model. Hitherto, security analyses based on STAMP framework have focused on safety analysis with cybersecurity causal factors.
  • This is (to the best of our knowledge) the only application of STAMP/STPA-Sec/Cybersafety on an industrial chiller plant which is an archetypal industrial control system.    

For the convenience of reviewers, we list each comment and detail our revision. In addition, we proof-read the entire paper to eliminate typographical errors.

Looking forward to hearing back from you. 

Yours,

Shaharyar Khan

On behalf of Stuart Madnick

---------------------

# Reviewer 3

-----------------

  • How does this approach compare with other related approaches?

Response 10: In this version of the paper, we have clarified the scientific contribution of this work. Specifically, the primary contribution of this work is to present an integrated safety/security model for an industrial control system application. While others claim to perform security analyses based on the STAMP framework, we argue that those are essentially safety analyses that identify cybersecurity causal factors. In this work, we create an integrated model by including security-specific requirements at every level of the analysis. To elucidate this point further, consider the controls in place when a security incident happens. The traditional STAMP methods would look at the PLC, operator etc., and determine how each of these could be causal factors for the attack to succeed. We are saying that in addition to that, we also need to look at security controls in place (their roles, responsibilities and actions) in addition to safety controllers/decision-makers. In some instances, the traditional safety controllers could have security responsibilities as well. In our analysis we explicitly identify security responsibilities for each controller.   

In addition, in this version of the paper, we have also completely revised the literature review section and presented a brief comparison of the various works/methodologies related to the proposed solution. We have also transformed the literature review section into a standalone section as advised by Reviewer #1.

  • The paper appears to be generally well written. There are some occasional typos (for example line 105: "safety control structure for the syste").

Response 11: We performed a spellcheck and fixed typographical errors in the manuscript.

Round 2

Reviewer 1 Report

The authors took all my comments into consideration and made the changes.